

# Uncertainties in cloud phase and optical thickness retrievals from the Earth Polychromatic Imaging Camera (EPIC)

Kerry Meyer[1,2], Yuekui Yang[1,2], Steven Platnick[2]

[1]Goddard Earth Sciences Technology and Research (GESTAR), Universities Space Research Association, Columbia, Maryland, USA

[2]NASA Goddard Space Flight Center, Greenbelt, Maryland, USA

*Correspondence to*: K. Meyer (kerry.meyer@nasa.gov)

**Abstract.** This paper presents an investigation of the expected uncertainties of a single channel cloud optical thickness (COT) retrieval technique, as well as a simple cloud temperature threshold based thermodynamic phase approach, in support of the Deep Space Climate Observatory (DSCOVR) mission. DSCOVR cloud products will be derived from Earth Polychromatic Imaging Camera (EPIC) observations in the ultraviolet and visible spectra. Since EPIC is not equipped with a spectral channel in the shortwave or mid-wave infrared that is sensitive to cloud effective radius (CER), COT will be inferred from a single visible channel with the assumption of appropriate CER values for liquid and ice phase clouds. One month of Aqua MODIS daytime granules from April 2005 is selected for investigating cloud phase sensitivity, and a subset of these granules that has similar EPIC sun-view geometry is selected for investigating COT uncertainties. EPIC COT retrievals are simulated with the same algorithm as the operational MODIS cloud products (MOD06), except using fixed phase-dependent CER values. Uncertainty estimates are derived by comparing the single channel COT retrievals with the baseline bi-spectral MODIS retrievals. Results show that a single channel COT retrieval is feasible for EPIC. For ice clouds, single channel retrieval errors are minimal ($< 2\%$) due to the particle size insensitivity of the assumed ice crystal (i.e., severely roughened aggregate of hexagonal columns) scattering properties at visible wavelengths, while for liquid clouds the error is mostly limited to within 10%, although for thin clouds (COT $< 2$) the error can be higher. Potential uncertainties in EPIC cloud masking and cloud temperature retrievals are not considered in this study.

## 1 Introduction

The Deep Space Climate Observatory (DSCOVR) satellite was launched on 11 February 2015, and in June 2015 began making observations of both the Earth and the Sun from its Lissajous orbit about the Earth's L1 Lagrangian point, a gravity neutral position 1.5 million km from the Earth. The DSCOVR payload includes two Earth-observing instruments, namely the Earth Polychromatic Imaging Camera (EPIC) and the National Institute of Standards and Technology (NIST) Advanced Radiometer (NISTAR). At the L1 point,





DSCOVR will remain near the Sun-Earth line, viewing the entire sunlit half of the Earth continuously at near backscatter directions with scattering angles ranging from roughly 164° to 176°. NISTAR observes the Earth as a single pixel, and provides measurements of the solar radiation reflected by the Earth, as well as the Earth's total radiant power, in four broadband channels. EPIC, with its $2048 \times 2048$ CCD (charge-coupled device) array, provides higher-spatial resolution details of the daytime Earth that can be used to interpret the single-pixel NISTAR data. Note that at the L1 point, the CCD array resolution yields roughly 8 km pixel sampling at the Earth's surface at nadir, though aggregating the native radiances to coarser resolution (e.g., $1024 \times 1024$) remains a possibility.

Numerous geophysical products will be derived from EPIC observations at 10 spectral channels ranging from the ultraviolet (UV) (317.5 nm) to the near-infrared (NIR) (779.5 nm) (Yang et al., 2013). These spectral channels are sensitive to various atmospheric and surface components, and can provide information on ozone, aerosol, cloud, and vegetation properties. The EPIC cloud products will include cloud masking, cloud optical thickness (COT), and cloud effective height. The EPIC cloud mask will adopt a threshold method utilizing information from multiple spectral channels; cloud height will be derived from the oxygen ($O_2$) A- and B-band observations (Yang et al., 2013). While the cloud mask and cloud height products are critical inputs for COT retrievals, this paper will focus solely on the EPIC cloud phase and COT algorithm and its potential uncertainties.

The EPIC COT product will be produced with the same core algorithms as those used by the operational MODerate-resolution Imaging Spectroradiometer (MODIS) cloud optical and microphysical property product (MOD06) (Platnick et al, 2003; Platnick et al., 2015) that is now part of a shared-core cloud retrieval algorithm suite that has been applied to other space-borne and airborne imagers such as the Visible Infrared Imaging Radiometer Suite (VIIRS) onboard the polar-orbiting Suomi-NPP (Platnick et al., 2013), the Spinning Enhanced Visible and InfraRed Imager (SEVIRI) onboard EUMETSAT's geostationary Meteosat (e.g., Hamman et al., 2014), and the Enhanced MODIS Airborne Simulator (eMAS) flown on NASA's ER-2 high-altitude research aircraft (e.g., King et al., 2010), among others. MOD06 provides cloud top pressure, temperature, and height retrievals from the infrared (IR) window and $CO_2$-slicing techniques (Menzel et al., 2008; Baum et al., 2012), as well as simultaneous retrievals of COT and cloud effective particle radius (CER) using the well known bi-spectral Nakajima and King (1990) method and cloud phase retrievals using a variety of IR brightness temperature, cloud top temperature, dual-phase CER, and spectral tests (Baum et al., 2012; Marchant et al., 2015). The MOD06 two-channel COT-CER retrieval approach couples a non-absorbing visible (VIS), NIR, or shortwave infrared (SWIR) spectral channel sensitive to COT with an absorbing SWIR or mid-wave infrared (MWIR) spectral channel sensitive to CER. However, because the EPIC spectral channels do not extend to wavelengths longer than the NIR, COT retrievals will be performed using a single-channel approach similar to that of the International Satellite Cloud Climatology Project (ISCCP) (Rossow and Schiffer, 1999) and the Multi-angle Imaging SpectroRadiometer (MISR) mission (Marchand et al., 2010), both of which assume appropriate values for CER. Since COT can be dependent on CER, in particular where the bi-spectral COT-CER retrieval solution



space is non-orthogonal such as at small COT, single-channel COT retrievals are thus prone to errors larger than those of the two-channel COT-CER retrievals.

A critical component of the COT retrieval process is determining the radiatively appropriate cloud thermodynamic phase. For EPIC, cloud thermodynamic phase will be inferred by imposing thresholds on cloud temperature. Due to the lack of thermal IR spectral channels, specifically those used for cloud altitude retrievals from either IR window or $CO_2$-slicing techniques, EPIC cloud temperature will instead be converted from $O_2$ A-band cloud effective height retrievals utilizing atmospheric profiles provided by the Goddard Earth Observing System Model, Version 5 (GEOS-5) (Rienecker et al., 2011). Oxygen, like $CO_2$, is a well-mixed gas, and reflectance measurements within the $O_2$ absorption region have been previously exploited for estimates of cloud altitude (e.g., Heidinger and Stephens, 2000; Rozanov and Kokhanovsky, 2004; Kokhanovsky et al., 2009). EPIC is equipped with two pairs of $O_2$ A-band (779.5 and 764 nm) and B-band (680 and 687.75 nm) reference and absorption channels that will be used to retrieve cloud effective height (Yang et al., 2013), which can then be converted to temperature and used to infer the cloud thermodynamic phase.

In this study, the expected EPIC COT retrieval uncertainties due to the required assumption of a fixed CER value will be estimated, in addition to the sensitivity of cloud thermodynamic phase retrievals to temperature thresholds. Moreover, the impact of EPIC's coarse spatial resolution, compared to the higher resolution polar-orbiting instruments such as MODIS (1 km resolution at nadir), to both cloud phase and COT retrievals and statistics will be explored. Proxy EPIC single-channel COT retrievals are obtained by running MOD06, assuming a fixed phase-dependent CER, on a subset of Aqua MODIS granules selected for its angular proximity to the backscatter region; expected COT errors are found via direct comparison with MOD06-like two-channel COT-CER retrievals. Cloud phase sensitivity is determined via adjustments to the cloud temperature threshold tests used for phase assignment. It should be noted that the error estimates provided here are intended to demonstrate whether a single channel COT retrieval will be feasible for EPIC. Furthermore, this study does not consider potential errors from sources such as cloud masking and cloud height retrievals, nor does it include 3D radiative transfer effects and sub-pixel cloud heterogeneity, though these error sources are expected to be substantial due to the large EPIC FOV (e.g., Davis et al., 1997).

## 2 EPIC COT Retrieval Algorithm Description

The planned operational EPIC COT product will leverage a shared-core cloud retrieval algorithm suite that includes the operational MODIS cloud optical and microphysical property product (MOD06) (Platnick et al, 2003). The shared-core concept provides consistency in retrieval methodology, ancillary datasets, etc., across multiple space-borne and airborne instrument platforms. Numerous enhancements to this retrieval suite were made for the recent MOD06 Collection 6 (C6) reprocessing effort (Platnick et al., 2015) and will thus be included in the EPIC retrievals. C6 enhancements relevant to EPIC include new bulk ice cloud



radiative models (Yang et al., 2013), improved characterization of surface reflectance for ocean (Cox and Munk, 1954a,b) and land (Schaaf et al., 2011), new cloud radiative transfer (RT) look-up tables (LUTs), and improved handling of ancillary datasets.

In addition, the C6 cloud optical property retrieval algorithm now processes and reports retrievals of pixels
that are identified as being only partially cloud covered, and are thus likely to deviate from the 1D homogeneous plane-parallel assumption. These "partly cloudy" (PCL) pixels, retrievals of which are reported separate from the "overcast" pixels, are expected to be poor retrieval candidates, and are generally associated with higher retrieval uncertainty and, for MODIS, increased rates of retrieval failure (i.e., the reflectance measurements lie outside the defined LUT solution space) (Cho et al., 2015). Identification of
these PCL pixels is accomplished via a clear sky restoral (CSR) algorithm (e.g., Zhang and Platnick, 2011; Pincus et al., 2012) that for MODIS uses sub-pixel cloud reflectance variability and cloud edge detection tests. For EPIC the cloud edge detection test will be applied, though the application of the sub-pixel spatial variability test depends on the availability of full-CCD resolution observations in at least one spectral channel. Because the EPIC pixel size is much larger than that of polar-orbiting imagers such as MODIS
and VIIRS, PCL pixels are expected to represent a larger fraction of the cloudy pixel population (Dey et al., 2008). Thus identifying these pixels is of increased importance, specifically as a means of assessing the quality of the optical property retrievals.

Since the EPIC channel set is limited compared to that of MODIS, there are important differences between the EPIC retrieval algorithm and its MOD06 counterpart. These include the use of a single channel,
680 nm, for COT retrievals with the assumption of fixed phase-dependent CER values due to EPIC's lack of absorbing spectral channels in the SWIR or MWIR that are sensitive to cloud particle size. The fixed CER values are derived from global MODIS retrieval statistics (e.g., King et al., 2013), and are assumed to be 12 µm and 30 µm for liquid and ice phase clouds, respectively. The lack of key SWIR or MWIR channels, for which liquid and ice absorption are known to differ (Kou et al., 1993), as well as IR channels
sensitive to cloud temperature/altitude, will also limit the available information content for determining cloud thermodynamic phase, i.e., liquid water, ice, or undetermined (i.e., ambiguous). Nevertheless, cloud phase can still be inferred by applying a dual threshold on cloud temperature derived from the $O_2$ A-band effective cloud height retrievals.

All ancillary datasets for EPIC COT retrievals will be identical to what are used by MOD06, with
appropriate substitutes when necessary. These include gap-filled land and snow/ice surface spectral albedos derived from MODIS (Schaaf et al., 2011), Near-real-time Ice and Snow Extent (NISE) (Nolin et al., 1998) data from the National Snow and Ice Data Center, and NCEP sea ice concentration (Grumbine, 1996, and references therein). For the operational EPIC retrievals, profiles of atmospheric temperature, pressure, water vapor, and ozone, as well as surface wind velocity, will be obtained from NASA's GEOS-5 model
(Rienecker et al., 2011); for the present investigation, however, these parameters will be obtained from the National Centers for Environmental Prediction (NCEP) Global Data Assimilation System (GDAS) 6-hour "Final Run" archive product (Derber et al., 1991) that is used in the current C6 MOD06 retrievals. The





forward-calculated COT retrieval look-up tables (LUTs) will be generated under assumptions and cloud models identical to those of MOD06 using the 1D plane-parallel discrete-ordinates radiative transfer (DISORT) method (Stamnes et al., 1988) ignoring atmospheric gaseous absorption, which is corrected for during the retrieval process; note that above-cloud Rayleigh scattering, expected to be non-negligible at

680 nm, will also be accounted for during the retrieval process on a pixel-by-pixel basis using the iterative technique of Wang and King (1997), similar to MOD06. For the forward calculated cloud retrieval LUTs, the liquid cloud droplet size distribution is assumed to be gamma distributed with effective variance of 0.1, with band-averaged scattering properties obtained by integrating spectral Mie calculations over the appropriate spectral response functions. Likewise, the ice crystal size distribution is also assumed to be

gamma distributed with effective variance of 0.1, with band-averaged scattering properties integrated over the spectral database of Yang et al. (2013); the ice clouds are assumed to be composed only of severely roughened compact aggregates of eight solid hexagonal columns (hereafter referred to as severely roughened aggregated columns), an assumption shown to provide better retrieval closure between solar-, IR-, and lidar-based retrievals of cirrus COT (Holz et al., 2015).

## 3. Methodology of Estimating EPIC Cloud Phase and COT Retrieval Errors

To investigate the expected EPIC cloud thermodynamic phase sensitivity, one month of Aqua MODIS daytime granules is selected for the analysis, namely April 2005. A subset of these granules that has angular coverage within the backward scattering angle region (164º to 176º) to be viewed by EPIC is selected for the COT retrieval error analysis. The use of one month of data ensures that a variety of liquid

and ice phase cloudy scenes are included, with sufficient sampling over both land and ocean surfaces. Figure 1 shows the geographic distribution of the counts of April 2005 Aqua MODIS pixels having scattering angles in the backscatter region ($\Theta > 164º$).

COT retrieval errors resulting from the fixed CER assumption are characterized using two versions of the MOD06 algorithm applied to the Aqua MODIS backscatter granule subset, specifically a baseline version

providing the full two-channel COT-CER retrievals and an EPIC proxy version assuming the fixed CER values of 12 and 30 µm for liquid and ice phase clouds, respectively. To simulate the EPIC 680 nm channel, both MOD06 versions use the 660 nm MODIS wavelength channel for COT retrievals.

For this study, cloud phase is determined by applying thresholds to the MOD06 1 km cloud top temperature (CTT) product, i.e., $CTT \leq 240\,K$ is assigned ice phase, $CTT \geq 260\,K$ is assigned liquid phase, and

$240\,K < CTT < 260\,K$ is assigned undetermined, i.e., ambiguous, phase. The error in cloud phase derived from this simple threshold method is determined by comparing with the operational MOD06 results for all April 2005 daytime granules (i.e., no backscatter angle filtering is applied). The sensitivity of this method to threshold selection is also investigated by perturbing the above-defined 1 km CTT thresholds by discrete temperature changes of increasing magnitude (e.g., $\pm 1\,K$, $\pm 2\,K$, etc.) and examining the resulting cloud



phase fraction changes. For all retrievals, pixel-level cloud masking information is obtained from the operational MODIS cloud mask product (MOD35) (Ackerman et al., 2010).

Note that for the operational EPIC COT retrieval, the above thresholds for cloud thermodynamic phase determination will necessarily be adjusted. As previously mentioned, the EPIC cloud temperature will be

derived from the $O_2$ A- and B-band effective cloud height retrievals. Unlike MOD06, for which cloud top pressure is determined using information from either the IR window or $CO_2$ channels (Menzel et al., 2008; Baum et al., 2012) that are more sensitive to the top of the cloud, the EPIC effective cloud height is more sensitive to the middle of the cloud (Yang et al., 2013). The operational temperature thresholds for EPIC cloud thermodynamic phase will be derived by collocating EPIC and MODIS observations, and using the

MOD06 phase results as the baseline once the EPIC cloud height product is available. We note again that this paper focuses on the uncertainties in COT retrievals resulting from a fixed CER assumption and the uncertainties in cloud thermodynamic phase determination using only cloud temperature thresholds; uncertainties in cloud temperature due to effective height retrieval errors are beyond the scope of this paper and are left for future investigations.

**4 Results**

**4.1 Cloud Thermodynamic Phase**

Because liquid and ice clouds have different radiative properties, determining the appropriate thermodynamic phase of a cloudy pixel is an important component of the EPIC cloud products. Figure 2 provides a case study example of the impact to cloud phase determination due to using only cloud

temperature thresholds, as described in Section 3, versus using a suite of spectral tests as implemented in MOD06. Here, cloud thermodynamic phase results from the operational MOD06 cloud optical properties phase product (c) and the proxy EPIC algorithm (d) that uses dual CTT thresholds (Section 3) are shown for an Aqua MODIS granule obtained over the eastern Pacific Ocean on 9 April 2005 (2215 UTC); the native MODIS resolution (1 km at nadir) is maintained in Fig. 2 and onwards until Section 4.3. Note the

MOD06 phase uses a variety of IR brightness temperature, cloud top temperature, dual-phase CER, and SWIR spectral tests (Baum et al., 2012; Platnick et al., 2015; Marchant et al., 2015). It is evident that limiting the information content to cloud temperature alone results in a significantly larger fraction of undetermined phase results (yellow shades), pixels for which additional spectral information can enhance the ability to discriminate between ice (light blue shades) and liquid water (dark blue shades) phases.

This increase in undetermined phase results is also seen in monthly statistics. Figure 3 shows a monthly cloud phase skill table comparing pixel-level phase results from the proxy EPIC algorithm to those from the MOD06 cloud optical property phase product. Data shown here are from all daytime April 2005 Aqua MODIS granules. Each box of the skill table indicates the fraction of cloudy pixels having the designated proxy EPIC (rows) or MOD06 (columns) cloud phase. For instance, the center box indicates that for 46.6%

of the cloudy pixels in this granule subset, the proxy EPIC and MOD06 phase results agree on liquid phase.



Note that the total proxy EPIC phase fractions can be found by summing each row, and likewise the MOD06 phase fractions by summing the columns; the sum of all boxes is 100%. Evidently the increase in undetermined phase results for the proxy EPIC algorithm, representing 22.2% of the cloudy pixel population, is at the expense of both the liquid and ice phase results, both of which represent smaller

fractions (48.2% and 29.6% for liquid and ice, respectively) than the respective MOD06 phase results (58% and 39.4% for liquid and ice, respectively). An increase in the number of undetermined phase pixels is not necessarily detrimental, however, since it likely prevents a sizeable number of otherwise incorrect liquid or ice phase results that can negatively affect COT retrieval quality.

In spite of the expected larger undetermined fraction due to EPIC's lack of phase information content, Fig.

3 shows that an algorithm using only cloud temperature thresholds agrees with the multiple-test MOD06 algorithm for roughly 77% of the global cloudy pixels in this granule subset (if the undetermined pixels are excluded, only 2.5% of the pixels are classified differently between the MODIS and EPIC algorithms). This phase agreement fraction, defined as the sum of the diagonal of the skill table in Fig. 3, can also be calculated on a global grid, as shown in Fig. 4. Here, the April 2005 phase agreement, normalized by 100

such that it represents a fraction of cloudy pixels within each grid box, is calculated on a 10° equal-angle grid. Evidently, the phase agreement is zonally dependent, with the highest agreement primarily in the tropics. This result implies that zonally dependent cloud temperature thresholds may be necessary for the operational EPIC cloud phase product; further investigation of the feasibility of such thresholds is left for future efforts.

The sensitivity of the cloud phase results to cloud temperature threshold selection is shown in Fig. 5. Here, monthly global cloud phase fractions, i.e., the fraction of cloudy pixels that are identified as liquid water (red line), ice (blue line), and undetermined (green line) phases, are plotted as a function of the cloud temperature threshold perturbation (see Section 3). These phase fractions are again calculated from all April 2005 daytime Aqua MODIS granules, using the MOD06 1 km CTT product. The temperature

threshold perturbations range from -5 K to +5 K, or a total magnitude of 10 K. Note that a sensitivity study showed that the EPIC A-band is expected to be capable of detecting cloud height changes equivalent to about 1 to 2 K, which correspond to altitude changes of roughly 150 m in the US standard atmospheric profile (Yang et al, 2013). As seen in the figure, increasing the CTT thresholds unsurprisingly yields a larger fraction of ice clouds and a smaller fraction of liquid water clouds; the fraction of undetermined, or

ambiguous, phase clouds also increases. Evidently, for a 10 K change in CTT thresholds, the liquid phase fraction decreases from roughly 55% to 41%, while the ice and undetermined phase fractions increase from roughly 26% to 34% and 19% to 25%, respectively. Moreover, these sensitivities appear to be nearly linear for all three phases, at least for the range of thresholds applied here. It should be noted, however, that the linear sensitivity shown here might become non-linear if an inappropriate cloud temperature threshold

range is selected. While the thresholds used in this analysis have been shown to be appropriate for CTT via analysis of CALIOP (Cloud-Aerosol Lidar with Orthogonal Polarization) observations (e.g., Marchant et





al., 2015), they will necessarily require adjustments for EPIC since cloud height information from the $O_2$ bands resides deep in the cloud as opposed to near cloud top (Yang et al., 2013).

**4.2 Cloud Optical Properties**

While cloud reflectance in the VIS spectrum is primarily dependent on the optical thickness of the cloud, there is also some sensitivity to particle size, specifically for liquid phase clouds that can be weakly absorbing in the VIS at larger droplet sizes, i.e., single scattering albedo becomes smaller than 1. Figure 6 shows simulated top-of-cloud 660 nm reflectance as a function of COT for liquid (red lines) and ice (blue lines) phase clouds. Here, solid lines denote the 12 μm and 30 μm fixed CER assumptions to be used for EPIC liquid and ice phase COT retrievals, respectively; dotted lines denote reflectance sensitivity to CER, specifically ±1 σ of reflectance assuming CER is uniformly distributed between 2-30 μm for liquid and 5-60 μm for ice. Reflectance is calculated using DISORT for solar and view zenith angles of 30°, and a relative azimuth angle of 172° (roughly 176° scattering angle); the clouds are assumed to overlie an ocean surface with a 7 m s⁻¹ surface wind speed. It is evident that, at least for the ice crystal habit assumed here (i.e., severely roughened aggregated columns), ice cloud reflectance at 660 nm is negligibly sensitive to CER. Liquid cloud reflectance at 660 nm, on the other hand, does exhibit sensitivity to particle size. Note in the real atmosphere the probability distribution function (PDF) of CER is not uniform as is assumed here, thus this figure likely overestimates the true CER sensitivity.

Returning to the Aqua MODIS granule in Fig. 2 (9 April 2005, 2215 UTC), Fig. 7 (a) shows the corresponding single channel COT retrievals that assume the fixed 12 and 30 μm CER for liquid and ice phase clouds, respectively. A dual color bar differentiates cloud thermodynamic phase, with liquid having warm colors and ice having cool colors. Cloud phase is determined here by the dual-threshold cloud temperature test described above and, following the convention in MOD06, undetermined phase is processed and plotted as liquid phase. The distribution of relative COT retrieval differences (i.e., errors) with respect to the baseline two-channel COT-CER retrieval is shown in (b) for liquid (red line) and ice (blue line) phase clouds. Note that the retrieval errors are only for the pixel population having scattering angle greater than 164° (see Fig. 2 (b)), roughly the expected EPIC scattering angle space. Consistent with Fig. 6, only minimal errors, 2% or less, are observed between the single channel fixed-CER and the baseline two-channel ice phase COT retrievals, while much larger errors, up to 10% or higher, are observed for liquid phase clouds.

On a global monthly scale, COT retrieval errors are similarly distributed. Figure 8 shows the distribution of relative single channel COT retrieval errors for liquid (red lines) and ice (blue lines) phases with respect to the baseline two-channel COT-CER retrievals for the April 2005 Aqua MODIS backscatter pixel subset shown in Fig. 1, i.e., pixels having scattering angle greater than 164°. Retrievals over all surfaces, land, and ocean are plotted as solid, dotted, and dashed lines, respectively. As in Fig. 7, the ice phase COT retrievals exhibit little sensitivity to CER, while the liquid phase retrievals are quite sensitive, with differences again





up to 10% or higher. Note also that the distribution of liquid phase retrieval differences is broader than in the granule example in Fig. 7.

Mean single channel COT retrieval errors (i.e., relative difference or bias) due to the fixed CER assumption are shown in Fig. 9 (a) as a function of retrieved COT and scattering angle for liquid (left) and ice (right) phase clouds for the April 2005 Aqua MODIS backscatter pixel subset in Fig. 1; also shown is single channel COT retrieval uncertainty (b). Here, COT retrieval error and uncertainty are calculated as the mean and standard deviation, respectively, of the histograms in Fig. 9. The black lines denote the corresponding COT retrieval PDFs. It is evident here that, as shown in Fig. 7 and 8, ice phase COT retrieval errors and uncertainty due to the fixed CER assumption are minimal, while the largest liquid phase errors and uncertainty correspond to small COT retrievals (primarily COT < 2) at the largest scattering angles ($\Theta > {\sim}174^0$); this is more easily seen by the magnified liquid phase plots (for COT < 10) in Fig. 10.

Finally, it is reasonable to expect that the global EPIC COT retrieval statistics can be dependent on the liquid, ice, and undetermined phase pixel populations identified by the dual-threshold cloud temperature phase discrimination approach. Table 1 shows the mean, median, and standard deviation of retrieved liquid and ice phase COT under different cloud phase scenarios, namely the multiple-test MOD06 phase, the dual-threshold EPIC phase, and the EPIC phase using CTT thresholds perturbed by ±5 K. Data are from the April 2005 backscatter pixel subset shown in Fig. 1, and include all cloudy pixels regardless of cloud phase agreement between scenarios; note that the cloud mask is identical for each scenario, thus the population of cloudy pixels is constant. It is evident that for this pixel subset, the ice phase COT retrieval statistics are relatively unaffected by cloud phase population differences, while the liquid phase COT retrieval means can vary by up to 1.5 COT.

### 4.3 Sensitivity to Spatial Resolution

Thus far, all expected EPIC retrieval errors and uncertainties have been estimated at native 1 km (at nadir) MODIS spatial resolution. However, because the nominal EPIC spatial resolution may be over 20 km if the native-resolution CCD radiances are aggregated to coarser resolution and, if necessary, after accounting for appropriate sensor spatial response functions, it is useful to characterize the effects of a relatively coarse spatial resolution on the cloud optical retrievals and their statistics. To this end, the modified MOD06 algorithm used to produce the proxy EPIC retrievals is further modified to aggregate 1 km MODIS pixels into 25 × 25-pixel boxes prior to the retrieval process, yielding cloud thermodynamic phase and COT retrievals at a spatial resolution of 25 km at nadir, considered to be a worst-case scenario for EPIC. Specifically, the 25 × 25-pixel aggregation involves averaging the measured 1 km spectral reflectance as well as the 1 km MOD06 CTT and CTP retrievals; note the MOD06 CT product is produced by an algorithm independent from the cloud optical retrievals and is not included in the shared-core retrieval suite used here. In addition, because the cloud masking is obtained from the 1 km MOD35 product, the 25 km EPIC-like "pixels" are deemed to be cloudy if the sub-pixel 1 km cloud fraction is greater than 16%, the same cloudiness threshold used in the operational MOD08 global level-3 aggregations of the MOD06 5 km




CT and IR cloud phase products (Hubanks et al., 2015). Finally, because the impacts of spatial resolution alone are of interest here, no scattering angle filtering is applied to the retrievals, i.e., all daytime April 2005 Aqua MODIS granules are used.

Figure 11 shows the 25 km resolution cloud phase (a) and single channel COT (b) retrievals for the Aqua MODIS case study granule shown in Fig. 2 and 7 (9 April 2005, 2215 UTC). Intuitively, the coarse resolution retrievals are unable to capture the fine-scale cloud horizontal structure that is evident in the 1 km phase and COT retrievals in Fig. 2 (d) and Fig. 7 (a), respectively, resulting in a smoother COT retrieval field. In addition, cloudiness is overestimated in regions with more broken clouds, a not unexpected result (e.g., Zhao and Di Girolamo, 2007), and the edges of the ice clouds are also more likely to be identified as undetermined phase. Note that increasing the sub-pixel 1 km cloud fraction threshold can mitigate overestimated cloudiness in broken cloud fields; however, it is unclear what threshold value may be more appropriate given that the real sensitivity of EPIC cloud detection to sub-pixel cloudiness remains unknown.

The observed granule-level differences ultimately will result in differences in spatially and temporally aggregated retrieval statistics. For instance, histograms of liquid (red lines) and ice (blue lines) phase single channel COT retrievals over (a) all surfaces, (b) land, and (c) ocean for all daytime April 2005 Aqua MODIS granules are shown in Fig. 12. Here, solid and dotted lines denote retrievals at 1 km and 25 km spatial resolution, respectively. As previously stated, the 25 km pixels must have 1 km sub-pixel cloud fraction greater than 16% to be included here; in addition, all cloudy 1 km pixels within each 25 km pixel must have cloud phase identical to that of the 25 km pixel. For liquid phase clouds, the smoothing of the COT retrieval field and overestimated cloudiness in broken cloud regions seen in Fig. 11 both contribute to 25 km COT retrieval distributions having smaller modes (~1 for all surfaces) than the 1 km COT distributions (~3 for all surfaces). The opposite is the case for ice clouds, as the modes of the 25 km COT retrieval distributions are larger (though only slightly) than those of the 1 km COT distributions.

## 4 Discussion and Summary

At the L1 point, the DSCOVR platform will continuously observe the entire sunlit half of the Earth at near backscatter directions, providing a unique perspective of the Earth's atmosphere, clouds, and aerosols. The DSCOVR cloud products will be derived from EPIC observations in the ultraviolet and visible spectra, with cloud optical property retrieval heritage from ISCCP and the operational MODIS cloud products (MOD06). Because a given location at the Earth's surface will be observed multiple times each day, this dataset is expected to provide insight into the cloud diurnal cycle. In addition, because EPIC is equipped with a 2048 × 2048 CCD array, the DSCOVR cloud products are expected to provide sub-pixel cloud information that can be used for interpretation of broadband solar reflectance data from NISTAR, which observes the Earth as a single pixel. Moreover, the DSCOVR cloud products can be complementary to other EPIC-based datasets such as UV aerosol and trace gas retrievals.



Due to the lack of CER sensitive channels, EPIC COT retrievals will adopt a single visible channel technique, and assume appropriate phase-dependent values for CER. The uncertainties resulting from this assumption are investigated using a subset of one month of Aqua MODIS granules from April 2005 for which MODIS observed the same scattering angle space as that observed by EPIC, i.e., the backscatter

angles between 164° and 176°. Results show that for ice clouds, the errors in the single channel retrieval are minimal ($<2\%$), while for liquid clouds the error can reach 10% or higher for thin clouds (COT $< 2$) at large scattering angles ($\Theta > 174°$).

EPIC will only have cloud temperature information from its $O_2$ A- and B-bands; hence for this study, cloud thermodynamic phase is determined using cloud temperature thresholds only. Compared to the operational

MOD06 cloud optical property phase results that use a suite of tests based on IR cloud top and spectral SWIR-based CER retrievals (Marchant et al., 2015), cloud phase determined from cloud temperature thresholds only is expected to yield more pixels with undetermined, or ambiguous, phase results. An investigation of the sensitivity of cloud phase determination to temperature thresholds show that a 10 K difference in thresholds results in absolute phase fraction changes of up to 14%, though it should be noted

that this sensitivity necessarily assumes that the thresholds are appropriately defined with respect to the distribution of cloud phase versus cloud temperature.

Finally, we note that potential uncertainties in the EPIC cloud mask and cloud temperature retrievals are not considered in this study. Furthermore, the COT retrieval approach assumes that the pixels are overcast and homogeneous, consistent with the plane-parallel forward radiative transfer assumption. Errors resulting

from sub-pixel cloud heterogeneity and 3D radiative transfer effects are quite beyond the scope of this investigation, and are in practice difficult to quantify. They nevertheless can be expected to be sizeable given the relatively large size of the EPIC field of view. Cloud masking errors, i.e., erroneously determining a given FOV is cloudy or clear, are likewise difficult to quantify, much less define, and are thus ignored here.





**Acknowledgments**

The authors would like to thank the leadership team of the NASA component of the DSCOVR project for their support of the development of the EPIC science algorithms, in particular Alexander Marshak, as well as the continued MOD06 cloud retrieval algorithm development support of Galina Wind and Nandana

5    Amarasinghe. This research was supported by NASA grant NNX15AB51G (DSCOVR Earth Science Algorithms program managed by Richard Eckman, PI Yuekui Yang) and by the NASA Radiation Sciences Program. The Aqua MODIS level-1 and level-2 data used in this investigation are publicly available from the    NASA    Level    1    and    Atmosphere    Archive    and    Distribution    System    (LAADS) (http://ladsweb.nascom.nasa.gov).



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





Table 1. Liquid and ice phase COT retrieval statistics using different cloud thermodynamic phase algorithms.

| Cloud Phase | COT Mean | | COT Median | | COT Standard Deviation | |
|---|---|---|---|---|---|---|
| | Liquid | Ice | Liquid | Ice | Liquid | Ice |
| MOD06 Phase | 9.2 | 12.6 | 3.7 | 3.6 | 15.9 | 26.2 |
| EPIC Phase | 8.2 | 12.5 | 3.2 | 3.5 | 14.8 | 26.2 |
| EPIC Phase, CTT-5K | 8.5 | 12.6 | 3.3 | 3.4 | 15.5 | 26.5 |
| EPIC Phase, CTT+5K | 7.7 | 12.6 | 3.0 | 3.6 | 14.1 | 26.1 |



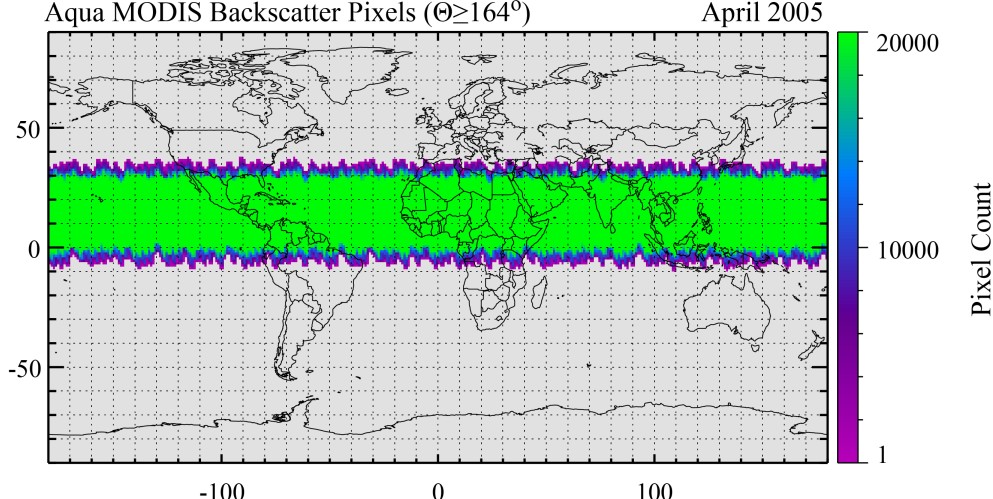

Figure 1. Geographic distribution of the April 2005 Aqua MODIS pixel subset having scattering angles in the backscattering region ($\Theta > 164°$).


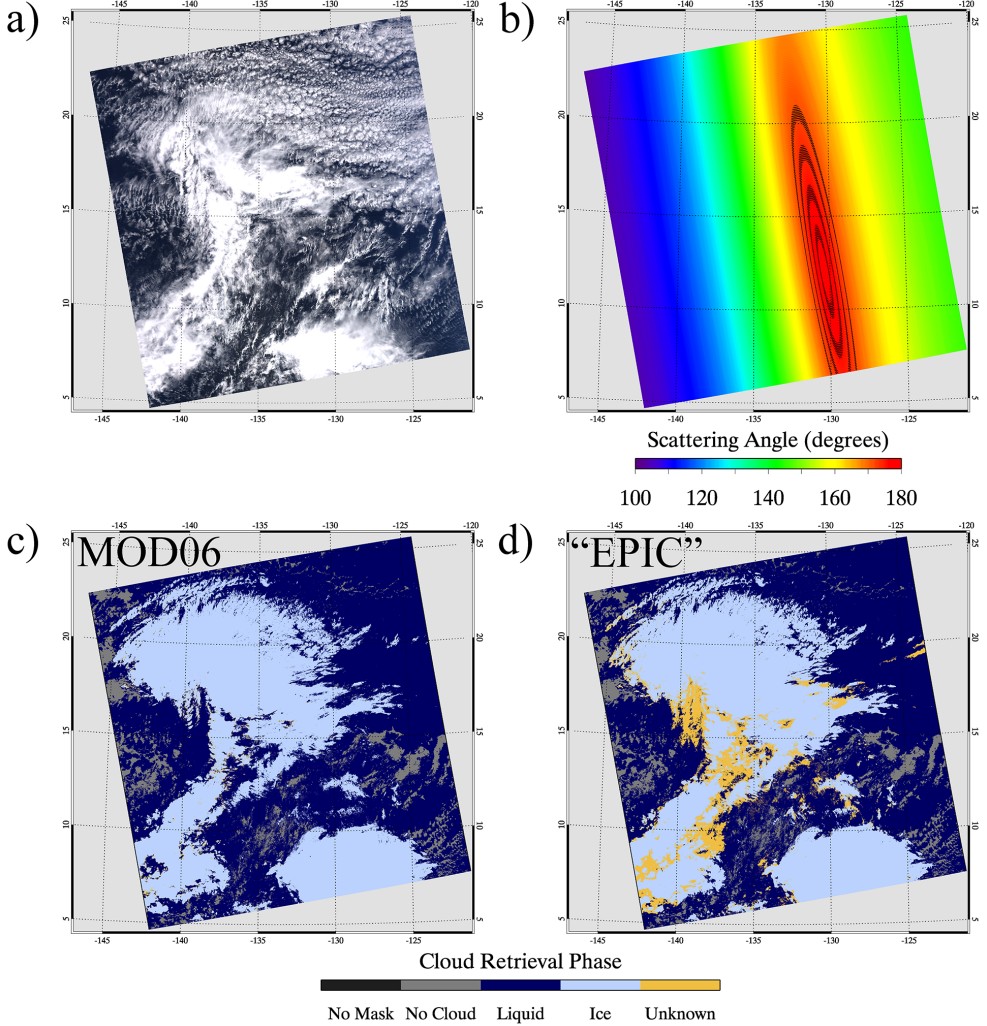

Figure 2. Example Aqua MODIS granule obtained over the eastern Pacific Ocean on 9 April 2005 (2215 UTC); the RGB image and corresponding scattering angle spatial distribution are shown in (a) and (b), respectively. This granule illustrates the cloud thermodynamic phase differences between the operational MOD06 cloud optical properties algorithm (c), which uses a variety of SWIR, IR and spectral CER tests, and an EPIC-style algorithm (d) using only cloud temperature thresholds.





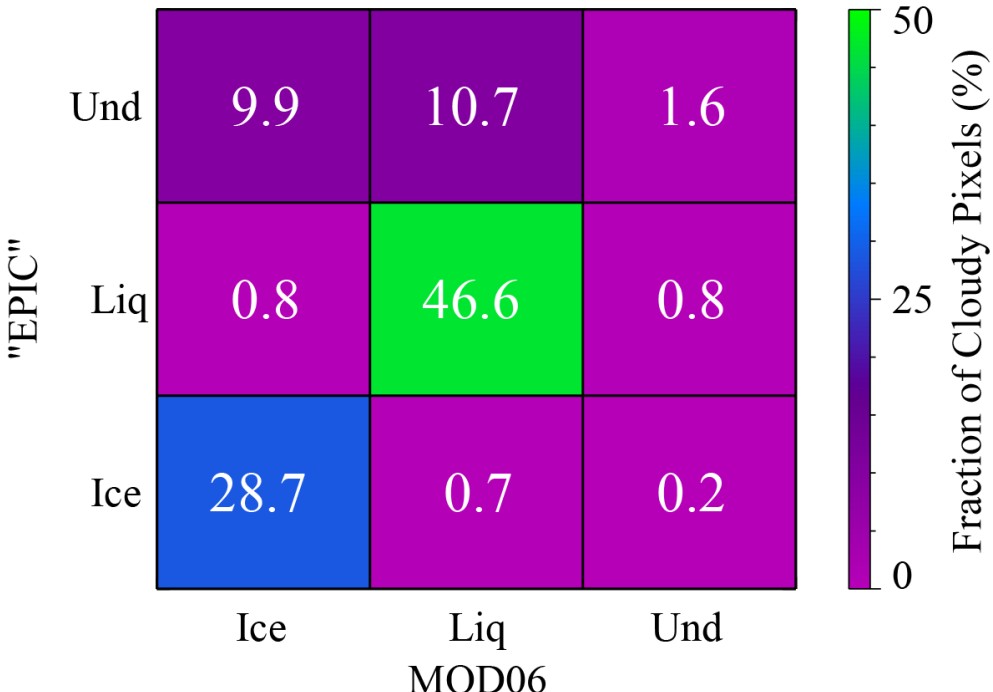

Figure 3. Monthly cloud thermodynamic phase skill table, from April 2005 Aqua MODIS, comparing the proxy EPIC algorithm results to those of MOD06. Although EPIC's lack of information content for phase will likely yield a larger fraction of undetermined phase results, an algorithm using only cloud temperature thresholds nevertheless agrees with the multiple-test MOD06 algorithm for roughly 77% of the global daytime cloudy pixels during this month.





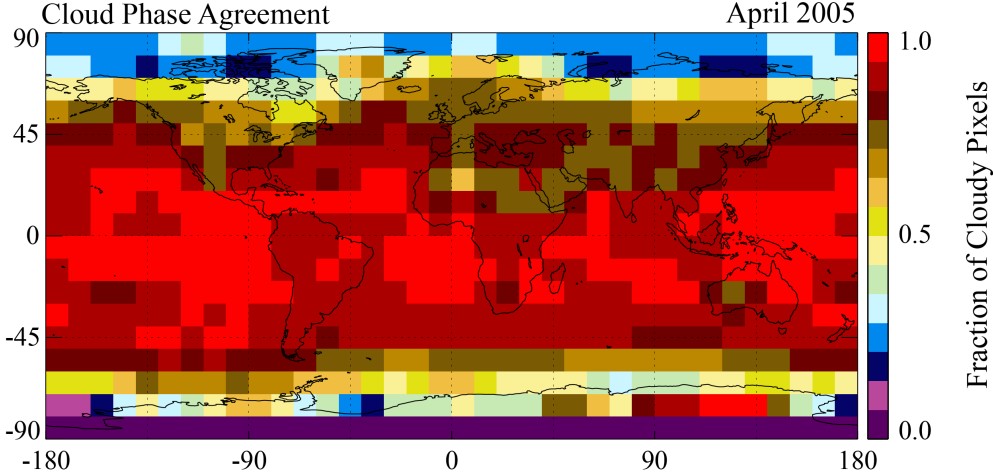

Figure 4. Gridded monthly cloud thermodynamic phase agreement fraction, defined as the fraction of cloudy pixels for which the proxy EPIC algorithm results agree with those from the multiple-test MOD06 algorithm, i.e. the gridded sum of the diagonal of the skill table in Figure 3.





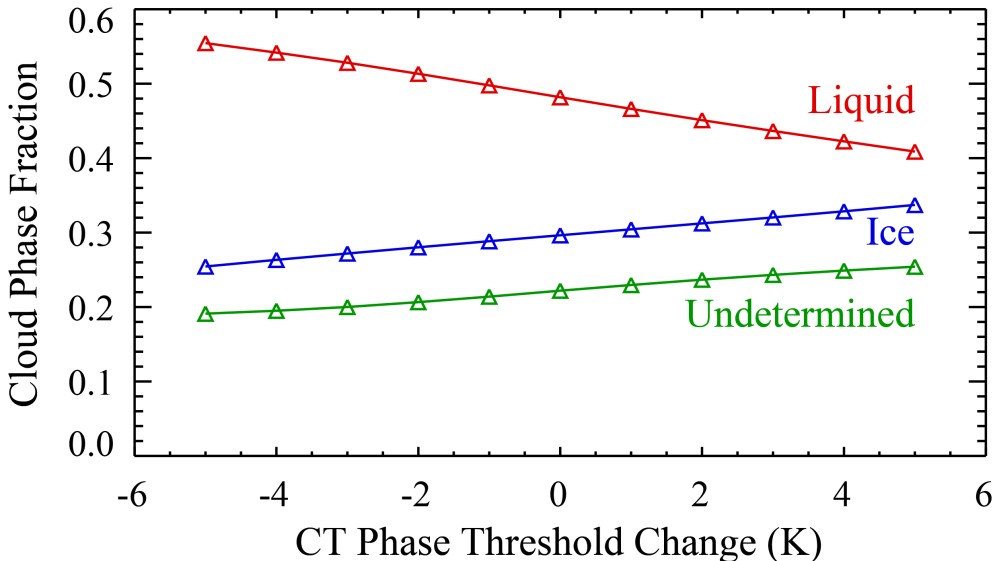

Figure 5. The expected sensitivity of the EPIC cloud thermodynamic phase algorithm, based only on cloud temperature thresholds, to errors in cloud temperature. Here monthly global cloud phase fractions, i.e., the fraction of cloudy pixels determined to be liquid (red line), ice (blue line), and undetermined (green line) phases, as a function of the CTT threshold perturbation are plotted. Data are from Aqua MODIS, April 2005.



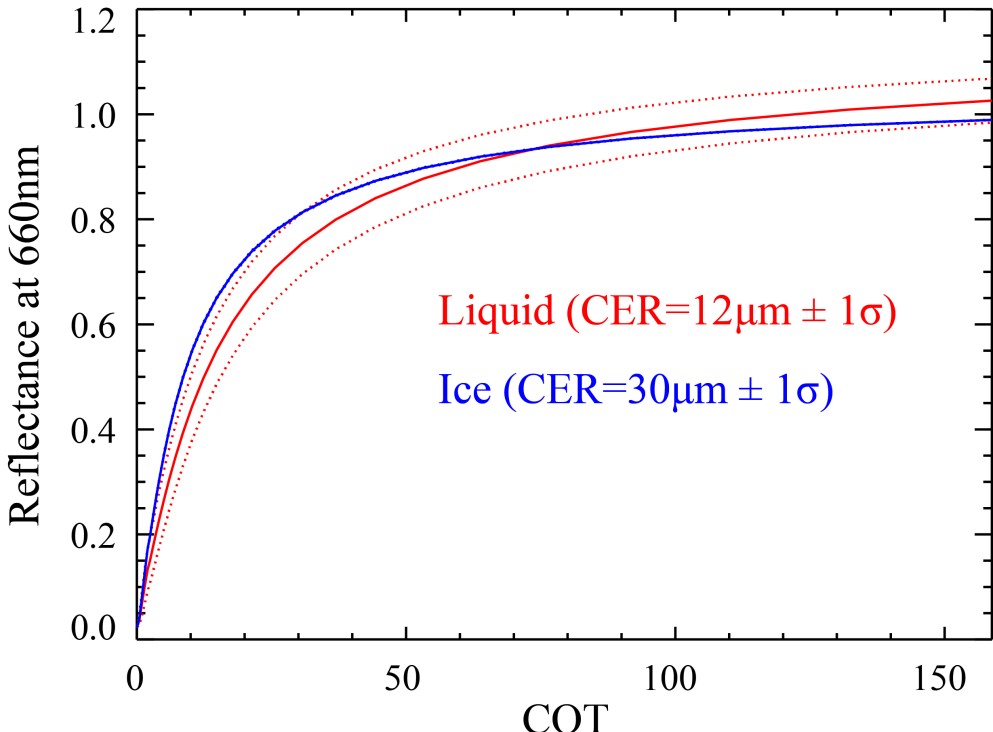

Figure 6. Sensitivity of cloud top 660 nm reflectance to CER, plotted as a function of COT. Solid lines denote reflectance for the phase-dependent CER assumptions for liquid (red, CER=12 µm) and ice (blue, CER=30 µm) phase. Dotted lines denote ±1σ of reflectance assuming CER is uniformly distributed between 2-30 µm for liquid and 5-60 µm for ice.



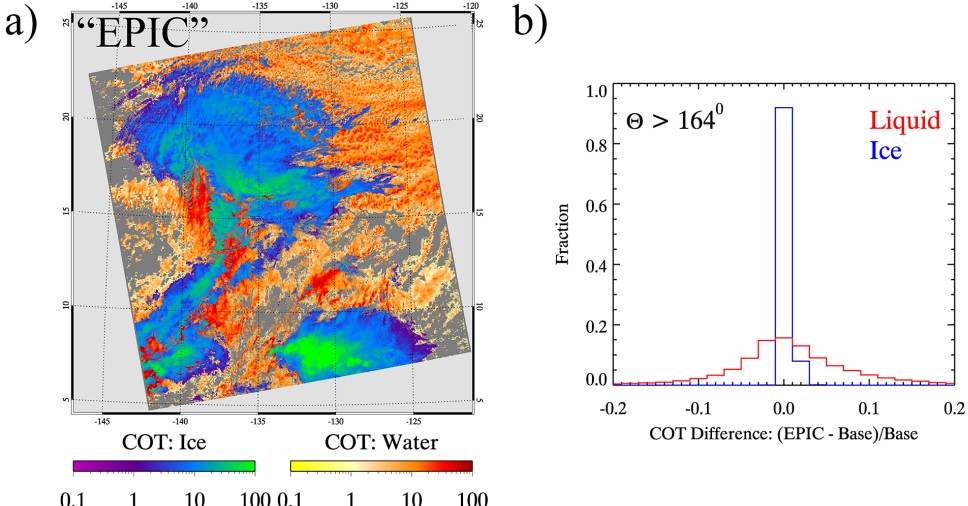

Figure 7. (a) Proxy EPIC COT retrievals corresponding to the Aqua MODIS granule in Figure 2, as well as
(b) the distribution of relative single channel COT retrieval differences (i.e., errors) with respect to the
baseline two-channel COT-CER retrievals for liquid (red line) and ice (blue line) phase clouds; the retrieval
errors are for the pixel subset having scattering angle greater than 164°.





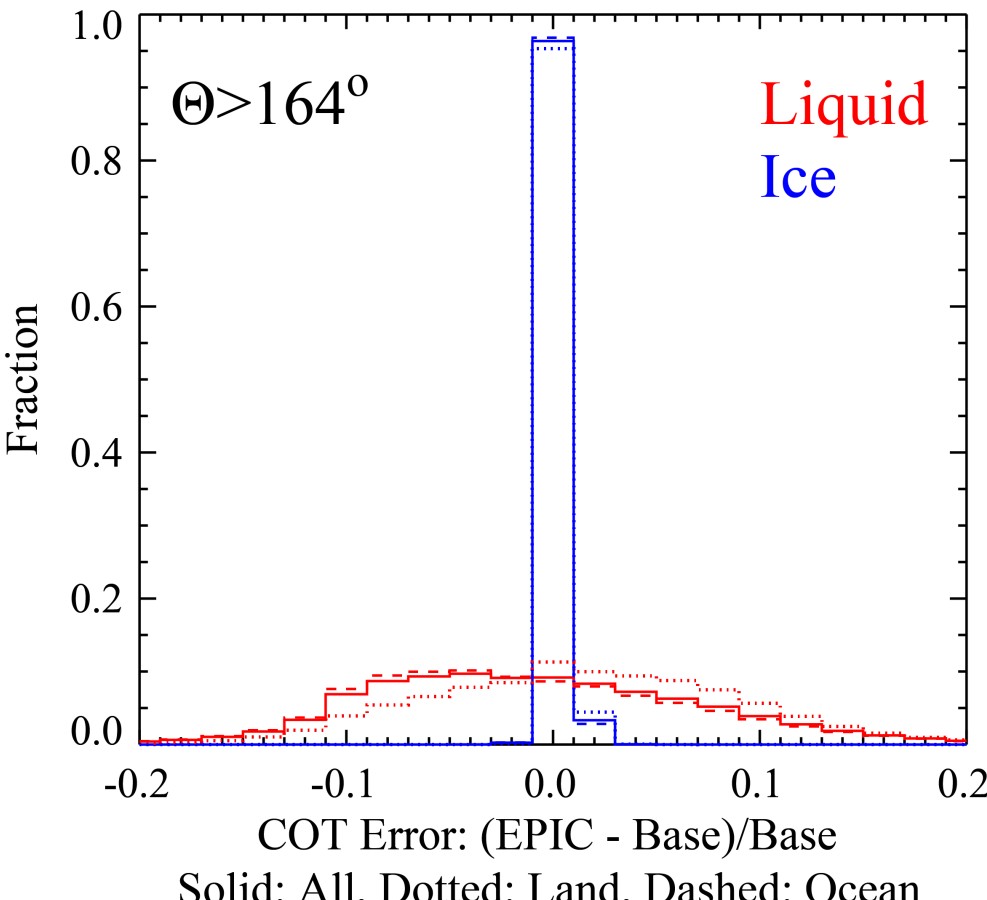

Figure 8. The distribution of relative single channel COT retrieval errors with respect to the baseline two-channel COT-CER retrievals for liquid (red lines) and ice (blue lines) phase clouds for the entire April 2005 Aqua MODIS granule subset. As in Figure 7 (b), the retrieval errors are for the pixel subset having scattering angle greater than 164° (see Figure 1). Retrievals over all surfaces, land, and ocean are plotted as solid, dotted, and dashed lines, respectively.





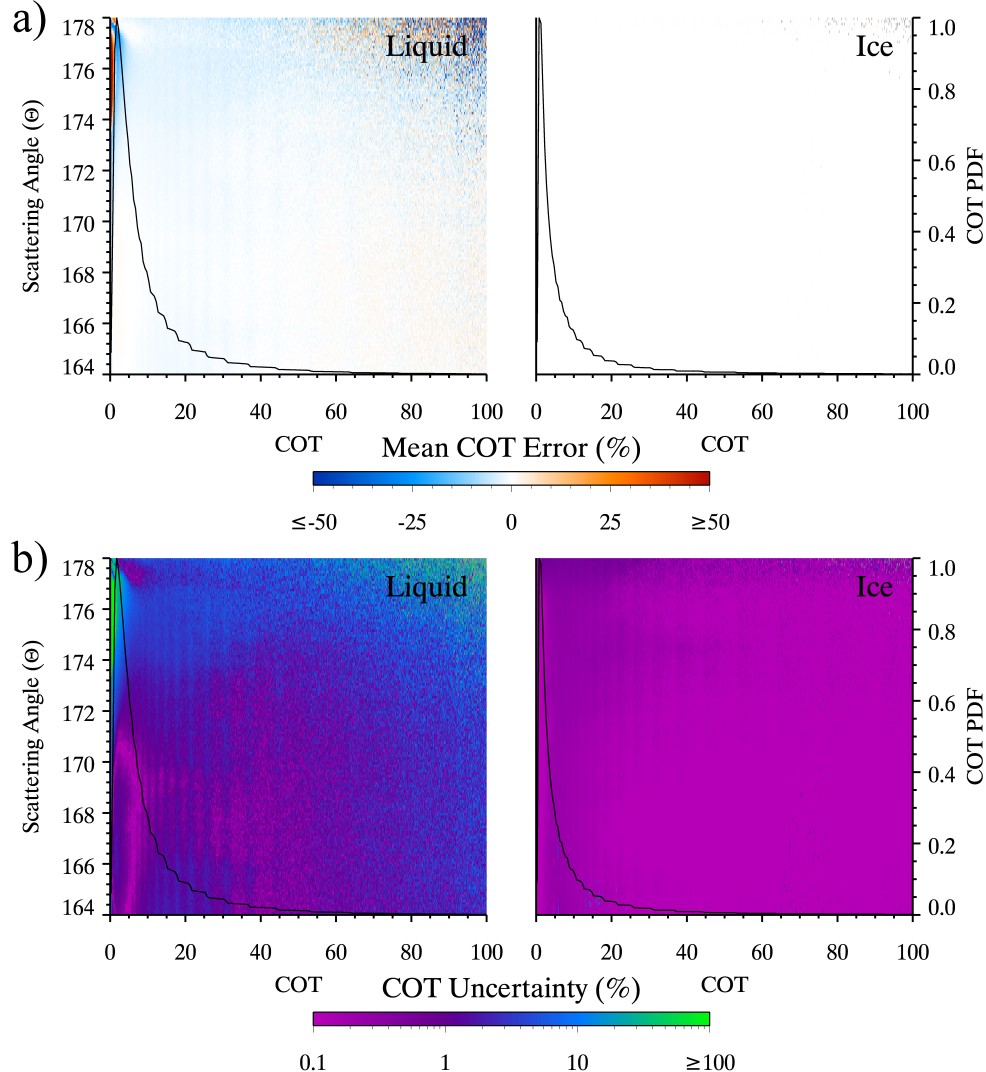

Figure 9. Mean single channel COT retrieval error (a) and uncertainty (b), with respect to the baseline two-channel COT-CER retrievals, as a function of COT and scattering angle for the entire April 2005 Aqua MODIS granule subset, limited to scattering angles greater than 164° (see Figure 1). COT retrieval error (i.e., relative difference or bias) and uncertainty are defined as the mean and standard deviation of the histograms in Figure 8. Black lines denote the COT retrieval PDFs.





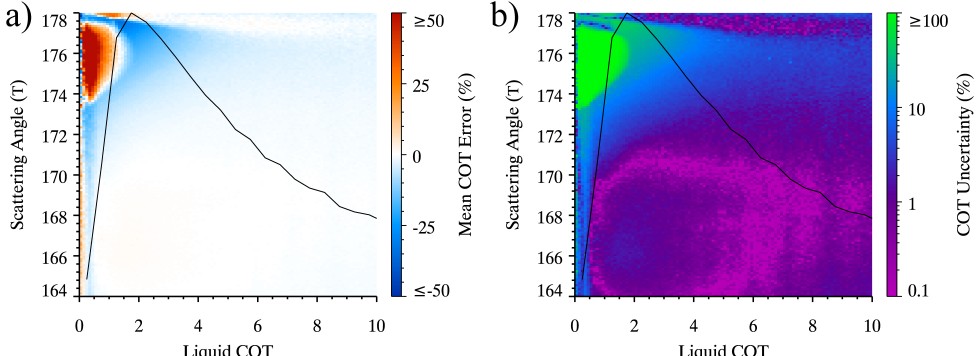

Figure 10. A magnified view of the mean single channel COT retrieval error (a) and uncertainty (b) corresponding to the liquid phase plots in Figure 9, here for COT < 10. Black lines again denote the COT retrieval PDFs.





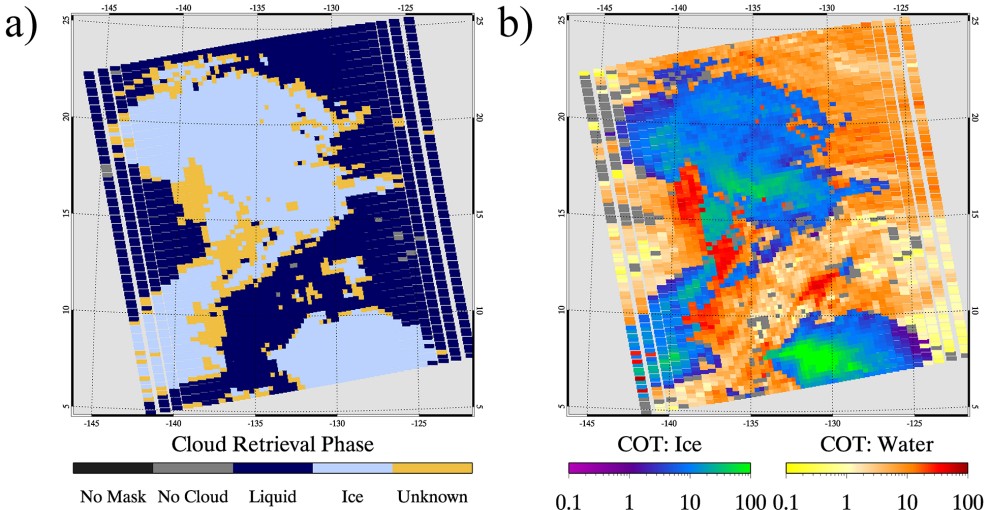

Figure 11. Cloud thermodynamic phase (a) and single-channel COT (b) retrievals corresponding to the Aqua MODIS granule in Figures 2 and 7. Here, the native 1 km (at nadir) MODIS resolution CTT and spectral reflectance are degraded to 25 km (at nadir) resolution prior to the retrieval process.





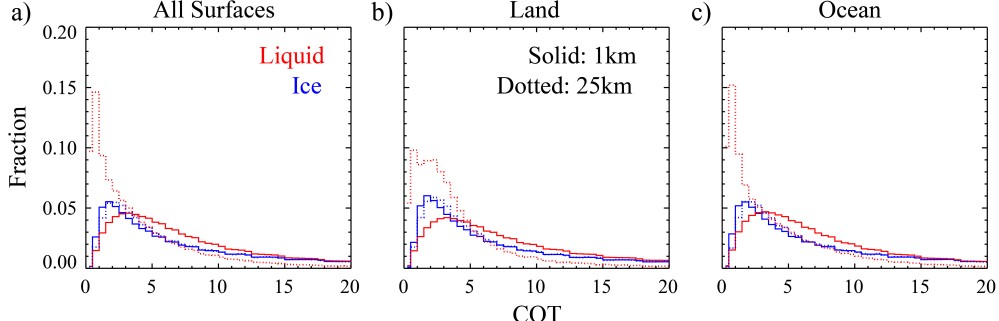

Figure 12. Histograms of single channel COT retrievals at the native 1 km (at nadir) MODIS resolution (solid lines) and 25 km (at nadir) resolution (dotted lines) for liquid (red lines) and ice (blue lines) phase clouds for the entire April 2005 Aqua MODIS subset. Note these histograms include only those scenes for which the cloudy 1 km pixels within each 25 km pixel have the same cloud phase as that of the 25 km pixel.