# Peer review of "Uncertainties in cloud phase and optical thickness retrievals from the Earth Polychromatic Imaging Camera (EPIC)"

_Atmospheric Measurement Techniques, 2016_

## Referee Comment (RC1) · R. Clay (Referee) · 2 Mar 2016

Does the paper address relevant scientific questions within the scope of AMT? Yes Does the paper present novel concepts, ideas, tools, or data? Yes Are substantial conclusions reached? Yes Are the scientific methods and assumptions valid and clearly outlined? Yes Are the results sufficient to support the interpretations and conclusions? On the whole Is the description of experiments and calculations sufficiently complete and precise to allow their reproduction by fellow scientists (traceability of results)? Yes Do the authors give proper credit to related work and clearly indicate their own new/original contribution? Yes Does the title clearly reflect the contents of the paper? Yes Does the abstract provide a concise and complete summary? Yes Is the over-

all presentation well structured and clear? On the whole Is the language fluent and precise? Yes Are mathematical formulae, symbols, abbreviations, and units correctly defined and used? On the whole Should any parts of the paper (text, formulae, figures, tables) be clarified, reduced, combined, or eliminated? Some - see below Are the number and quality of references appropriate? Yes

This paper deals with the expected uncertainties in retrieving cloud optical thickness and temperature from the Earth Polychromatic Imaging Camera (EPIC) in support of the Deep Space Climate Observatory mission. The camera has limitations which may affect those uncertainties and these are tested using MODIS data which allows a comparison between MODIS analyses and results which would have been obtained with the more limited EPIC data.

This paper is generally excellent in satisfying its apparent aims.

The data are analysed comprehensively and convincingly with limitations and degraded uncertainties broadly as one might expect. The arguments are clear and detailed and the presentation, though dense, is appropriate. The limitations of EPIC mean that some cloud parameters have to be assumed and the results of these assumptions are discussed in sufficient detail.

We are not told to what extent the uncertainties satisfy any predetermined design criteria but this paper will clearly be useful as a baseline document for later users of EPIC data.

I was perplexed to discover that (page 7 line 5) "An increase in the number of undetermined phase pixels is not necessarily detrimental..........". The following explanation may be correct but the sentiment could be better expressed.

Units are generally clear and consistent except in line 13 of page 11 we see degrees (presumably Celsius) replacing"K" as the temperature unit.

Acronyms are generally referenced and explained when introduced. DISORT, page 8,

seems to be an exception. It is fundamental, but the cloud optical thickness product (COT) must be at a particular wavelength. We should be clearly told what that is early on.

---

## Referee Comment (RC2) · Anonymous Referee #2 · 22 Mar 2016

This manuscript reports retrieval uncertainty in cloud phase and optical thickness from EPIC measurements, part of the DSCOVR mission. The authors use current MODIS cloud products as a baseline, and then discuss the sources of EPIC retrieval errors introduced by the limitation of channels (and thus the method itself), and by the coarser resolution of EPIC.

The manuscript is well structured and well written, although it could have been more concise (see specific comments below). The sensitivity tests are rather standard, but indeed it is important to report these results to provide a good reference for future operational cloud products. I only have a few minor comments for the authors to consider.

Specific comments:

[Figure]

1) While it is great to reinforce key differences between MODIS and EPIC, there are a lot of text repetitions in the manuscript. For example, the part about fixed cloud effective radius on Page 2, 4 and 6. It is also mentioned many times about why it is necessary to use dual thresholds for cloud phase determination, etc. I would recommend reading through the whole paper again, then the authors will realise that many same references are mentioned over and over again, indicating that some reorganisation could be made.

In addition, Page 7, Line 33-35, I find the sentence is interesting but not necessary, because why one would like to select an inappropriate cloud temperature threshold?

2) Page 9, Line 7–11: Could the authors please explain why large retrieval errors occur at certain scattering angles?

3) Page 7: it would have been better if the authors tried to implement zonally-dependent cloud temperature thresholds. Or at least, the author could analyse MODIS cloud products to support the temperature range used in Figure 5.

4) Regarding Figure 5 and Page 5, it is not immediately clear if both thresholds increase 1K , or the authors change one at a time. Please make it clear in the text and the figure caption.

5) Page 4: It is mentioned that different data sets are used for atmospheric profiles. Could the author please explain why, and elaborate on the potential impact on retrievals?

6) Page 5: It is not immediately clear if retrieval is performed at MODIS pixels or at EPIC pixels for "EPIC proxy version", although one can figure it out later when reading the result section. It would be good to make things a bit clear here.

---

## Author Comment (AC1) · 12 Apr 2016

The authors would like to thank the reviewer for the careful evaluation of our manuscript and constructive comments. Our responses to specific comments are below.

*We are not told to what extent the uncertainties satisfy any predetermined design criteria but this paper will clearly be useful as a baseline document for later users of EPIC data.*

We thank the reviewer for the kind words, and note that is difficult to identify any predetermined design criteria for EPIC, especially for clouds.

*I was perplexed to discover that (page 7 line 5) "An increase in the number of undetermined phase pixels is not necessarily detrimental..........". The following explanation may be correct but the sentiment could be better expressed.*

We agree with the reviewer that the sentiment could be better expressed; perhaps an adjective more appropriate than "detrimental" is "undesirable." We note that an incorrect identification of thermodynamic phase for a given cloudy pixel can not only result in a biased COT retrieval for the pixel itself, but can also introduce biases into global aggregations of retrieval statistics, particularly those statistics that are aggregated according to phase such as COT (e.g., mean liquid phase COT, etc.). Therefore, rather than forcing binary phase decisions (i.e., ice or liquid) for pixels for which the available information suggests ambiguous phase results, we leave these pixels in the undetermined category so as not to introduce biases into the aggregated statistics. We concede, however, that such an approach itself can cause a sampling bias if ambiguous/undetermined phase results are systematically associated with pixels having, for instance, a specific range of COT. We have modified the text to better express our reasoning (p. 7, lines 12-15).

*Units are generally clear and consistent except in line 13 of page 11 we see degrees (presumably Celsius) replacing"K" as the temperature unit.*

We thank the reviewer for identifying this inconsistency, and have replaced "degrees" with the correct unit "K" (p. 13, line 21)

*Acronyms are generally referenced and explained when introduced. DISORT, page 8, seems to be an exception.*

We note that DISORT was defined on page 5 (line 5) as being the "discrete-ordinates radiative transfer (DISORT) method." To eliminate any confusion, however, we have capitalized the relevant letters such that a clearer link between the acronym and full name can be made.

*It is fundamental, but the cloud optical thickness product (COT) must be at a particular wavelength. We should be clearly told what that is early on.*

We thank the reviewer for pointing this out. As is common practice in the cloud remote sensing community, COT here is referenced to the visible $0.66\,\mu m$ wavelength. We have included this specific detail in the Section 2 algorithm description (p. 4, lines 20-22).

---

## Author Comment (AC2) · 12 Apr 2016

The authors would like to thank the reviewer for the careful evaluation of our manuscript and constructive comments. Our responses to specific comments are below.

*1) While it is great to reinforce key differences between MODIS and EPIC, there are a lot of text repetitions in the manuscript. For example, the part about fixed cloud effective radius on Page 2, 4 and 6. It is also mentioned many times about why it is necessary to use dual thresholds for cloud phase determination, etc. I would recommend reading through the whole paper again, then the authors will realise that many same references are mentioned over and over again, indicating that some reorganisation could be made.*

> While we understand the reviewer's concern regarding repetition, we believe much of the repetition is necessary as a means to reinforce many of the salient assumptions and concepts of the current investigation. That said, we did move one sentence from the introduction (Section 1) to the algorithm description (Section 2) (see p. 4, lines 22-24).

*In addition, Page 7, Line 33-35, I find the sentence is interesting but not necessary, because why one would like to select an inappropriate cloud temperature threshold?*

> We appreciate the reviewer's question. The notion underlying this statement is not that one would deliberately select inappropriate cloud temperature thresholds (indeed, why would one like to!), but that the thresholds may not be appropriate for all cases. This is particularly relevant given the expected difficulty of applying thresholds based on the $O_2$ A-band cloud temperatures that are weighted more towards the middle of the cloud rather than towards cloud top. We in fact allude to this in the succeeding sentence (p. 8, lines 5-8).

*2) Page 9, Line 7–11: Could the authors please explain why large retrieval errors occur at certain scattering angles?*

> This is an excellent question. The large liquid COT retrieval errors at certain scattering angles (e.g., $174° < \Theta < 178°$) are almost certainly a result of the sensitivity of the scattering phase function at those angles to effective size. Note that these differences are largest at small COT ($<2$), where the single scattering component is expected to dominate the total reflectance. The figure below shows relative phase function differences, with respect to (and normalized by) the phase function of the assumed liquid $CER = 12 \mu m$, at the $0.66 \mu m$ wavelength for different liquid CER – blue and red shaded lines denote CER smaller and larger than $12 \mu m$, respectively. The largest phase function differences correspond to the $174° < \Theta < 178°$ scattering angle range, consistent with Figs. 9-10. We have added a statement to the manuscript that summarizes this phase function sensitivity (p. 9, lines 17-19).

[Figure]

*3) Page 7: it would have been better if the authors tried to implement zonally-dependent cloud temperature thresholds. Or at least, the author could analyse MODIS cloud products to support the temperature range used in Figure 5.*

This is an excellent comment, and we note that the results shown in Fig. 5 have indeed persuaded us to pursue a zonally-dependent cloud temperature threshold for the operational EPIC product. We also note that the temperature thresholds assumed here were in fact derived from a global analysis of one month (November 2012) of the MODIS cloud products and the active lidar products of CALIOP; see the figure below (Figure 11 from Marchant et al., 2016). That said, it is important to re-emphasize that the thresholds used here are appropriate for the IR-derived cloud temperature retrievals that are more sensitive to the top of the cloud compared to the $O_2$ A-band retrievals that are more sensitive to the middle of the cloud. Thus defining the appropriate zonal thresholds for use with the EPIC $O_2$ A-band cloud height product is left for future efforts, in particular after a sufficient amount of data has been produced such that a thorough analysis with co-located lidar observations can be pursued. We have added text clarifying the provenance of the CTT thresholds used here (p. 5, lines 33-35).

[Figure]

(Figure 11, Marchant et al., 2016). Probability density functions (PDFs) of CALIOP (a) and MODIS C6 (b) and C5 (c) cloud phase against the MODIS 1km cloud top temperature for November 2012.

*4) Regarding Figure 5 and Page 5, it is not immediately clear if both thresholds increase 1K , or the authors change one at a time. Please make it clear in the text and the figure caption.*

This is a good comment. We do shift both the liquid and ice thresholds simultaneously in the same direction, such that both either increase by the stated amount, or decrease by the stated amount. We have clarified this in the text (p. 6, line 5).

*5) Page 4: It is mentioned that different data sets are used for atmospheric profiles. Could the author please explain why, and elaborate on the potential impact on retrievals?*

This is another good comment. We first note that all retrievals shown in the manuscript use the NCEP GDAS ancillary profiles. For production of the official NASA EPIC dataset that will be available to the public, however, the NASA GEOS-5 ancillary profiles will be used per project requirements. We have opted for the NCEP GDAS profiles here simply because they are currently used for the operational NASA MODIS products (MOD06), and the required algorithm code is already in place; we note that the MOD06 algorithm team is currently testing the GEOS-5 profiles as a replacement for GDAS, though this functionality is not yet available to the current investigation. Nevertheless, for COT retrievals, the impact of the atmospheric profile is relatively minor since the spectral channel used for these

retrievals, the 680 nm channel that is the reference for the 687 nm $O_2$ B-band, is largely free of atmospheric absorption. The largest impact of the atmospheric profile assumption is expected to be on the $O_2$ A-band height retrievals, though we again emphasize that investigating the uncertainties of these retrievals is beyond the scope of this manuscript.

*6) Page 5: It is not immediately clear if retrieval is performed at MODIS pixels or at EPIC pixels for "EPIC proxy version", although one can figure it out later when reading the result section. It would be good to make things a bit clear here.*

This is a good suggestion. We have added text clarifying the retrieval resolution (p. 6, lines 5-7).